# The Profile, Health Seeking Behavior, Referral Patterns, and Outcome of Outborn Neonates Admitted to a District and Regional Hospital in the Upper West Region of Ghana: A Cross-Sectional Study

**DOI:** 10.3390/children7020015

**Published:** 2020-02-18

**Authors:** Edem M. A. Tette, Benjamin Demah Nuertey, Emmanuel A. Azusong, Naa Barnabas Gandau

**Affiliations:** 1Department of Community Health, University of Ghana Medical School, P.O. Box 4236, Accra, Ghana; 2Public Health Department, Tamale Teaching Hospital, P.O. Box, TL 16, Tamale, Ghana; 3Upper West Regional Hospital, P.O. Box 6, Wa, Ghana; azusongemmanuel@gmail.com (E.A.A.); naabarnabas@gmail.com (N.B.G.); 4School of Medical Science, University for Development Studies, Tamale, Ghana

**Keywords:** health seeking, socio-cultural, referral pattern, neonate, Sustainable Development Goals (SDGs), newborn

## Abstract

Neonatal mortality is the major contributor to under-five mortality rates in many low and middle income countries. We examined the health practices, care-seeking behavior, and referral of sick outborn neonates to a district and regional hospital in the Upper West Region of Ghana. The study was a cross-sectional study conducted over an eight (8) month period in 2018. Data were obtained from caregiver interviews and case notes. Altogether, 153 outborn neonates were examined. Inappropriate practices including the use of enemas, cord care with cow dung, and herbal baths were found. Three babies treated this way died. The majority of caregivers sought care at a health facility. However, 67 (44%) sought care only after their babies were ill for ≥7 days, suggesting the influence of a period of confinement on health seeking. More than half, 94 (61.4%), of the facilities visited referred patients to destination hospitals without giving any treatment. Delayed care-seeking was associated with a low birth weight, using home remedies, and a maternal age of ≥30 years. Altogether, 12 neonates (7.8%) died, consisting of three males and nine females (*p* = 0.018). Socio-cultural factors strongly influence health seeking behavior and the health outcome of neonates in this setting. There appeared to be a limited repertoire of interventions for treating neonatal disease in primary care.

## 1. Introduction

Neonatal mortality has assumed increasing prominence over the years and is now well-recognized as the major contributor to under-five mortality rates in many low and middle income countries [1,2,3]. Efforts to reduce the under-fives mortality rate and achieve the 4th Millennium Development Goal (MDG 4) only made modest gains in reducing the neonatal mortality rate [2,3]. Since the determinants of neonatal health tend to vary from one locality to another, these differences need to be identified and addressed in order to achieve the Sustainable Development Goals (SDGs) target to end preventable newborn deaths [1,2].

While efforts are being made to reduce neonatal mortality by enhancing the standard of neonatal care in hospitals, neonatal mortality, like maternal mortality, is also influenced by factors causing delay in reaching a hospital, which need similar attention [4]. This, according to the model used by Thadeus and Maine to describe delays contributing to maternal mortality, includes delays in taking the decision to seek health care and delays in reaching a hospital before the delays at the health facility level are eventually experienced. Unfortunately, many of the neonatal deaths occurring before the arrival at a hospital go unregistered and are not captured by routine data, particularly in rural communities where it matters most [5]. Unless the care barriers in these settings are identified and addressed, mortality may remain high within some localities of countries, such as Ghana, while aggregated figures show significant improvements in neonatal mortality rates. This was found to occur in Bangladesh following a review of two demographic and health surveys [6].

Several factors influence newborn health and the care seeking behaviors of their caregivers [5,7]. This ranges from the awareness of danger signs, essential newborn care practices, personal and communal beliefs, health and socio-cultural practices, as well as financial and physical access to health services [5,7,8,9,10,11]. The level of service that is readily available close to them can also affect health [9,10,12]. Thus, the robustness of the referral system, the type of care that is available, and the providers at each level, from home to hospital, are important determinants of neonatal outcome [1,11,12,13,14,15]. We examined the health practices, care seeking behavior, and selected aspects of the processes involved in the referral of outborn neonates to a district and a regional hospital in the Upper West Region of Ghana, the region with the most rural population during the last census in 2010, in order to identify care barriers in this and similar settings.

## 2. Materials and Methods

### 2.1. Study Area

The study was carried out at the Upper West Regional Hospital (UWRH) and St. Joseph’s Hospital (SJH). The Upper West Regional Hospital doubles as a municipal and regional hospital of people living in the Upper West Region of Ghana. It serves as the main referral center for the region and surrounding towns in the Northern, Upper East Region, and Burkina Faso, which shares a border with the region on its northern side. At the end of 2016, it had nine wards and a bed capacity of 200 beds, including a neonatal unit which was created between August and December 2016. Altogether, 248 neonates were admitted with 39 deaths in the same period. There are no intensive care services. The number of deliveries at the UWRH in 2016 was 4915.

St. Joseph’s Hospital in Jirapa functions as the district hospital for the people of Jirapa. It is one of the hospitals belonging to the Christian Health Association of Ghana (CHAG). The hospital had a bed capacity of 193 beds in early 2017 distributed over seven (7) wards. In 2016, the total deliveries were 1709. Until 2018 when a neonatal unit was created at the UWRH, it was the main referral center for neonatal conditions in the region. Patients from surrounding health centers, Community-Based Health Planning and Services (CHPS), compounds, and some of the district hospitals in the region were referred to the hospital. The UWRH and the St. Joseph’s Hospital are about 64.07 km apart. Figure 1 below shows the study area, towns, and health facilities. In addition, distance and access to transportation considerations influencing the choice of which referral center to go, and other factors, such as patient beliefs, past experience of family and friends, as well as the size and cadre of staff at the referral facilities, may help inform a patient to choose to which facility to send their neonate.

### 2.2. Study Design

The study was a descriptive cross-sectional study which examined the profile, health seeking behavior, and outcome of outborn neonates admitted to the Upper West Regional Hospital (UWRH) and St. Joseph’s Hospital (SJH) over an eight (8) month period from 23 January to 16 October 2018. The health seeking behavior of caregivers of neonates, the referral system, and sources of delay were determined.

### 2.3. Sampling and Sample Size

Parents or caregivers of all the 153 consecutive outborn neonates admitted over the study period were interviewed. Although the study collected data over an eight (8) month period from 23 January to 16 October 2018, the last outborn neonate from SJH was admitted in June 2019 resulting in the low numbers of outborn neonate from SJH. These estimates were calculated from the 2016–2017 total deliveries, neonatal admissions, and proportion of admitted neonates that were outborn in the study facilities. Outborn neonates were defined as neonates born outside the Upper West Regional Hospital (UWRH) or St. Joseph’s Hospital (SJH) and admitted to these hospitals during the study period. The estimated sample size of patients recruited for the maternal interviews was based on the assumptions that 15–20% of newborns required admission, and of these, approximately 25–60% were outborn babies. Thus, outborns will form around 10% of all newborns. Therefore, for a confidence interval of 95% and allowable error of 5%, the minimum sample size was calculated to be 138 patients. Thus, we aimed at obtaining 150 outborn caregiver interviews.

### 2.4. Study Population

Caregivers of newborns admitted to the neonatal unit who delivered outside the hospital in 2018 during the study period from 23 January to 16 October 2018 were eligible for this part of the study. Those who were unavailable to provide accurate information on the referral process or unable to provide consent were excluded from the study.

### 2.5. Data Collection

Data on parental health seeking behavior and the referral process were collected by administering questionnaires to caregivers and reviewing the child’s case notes. This data included information on the place the child was referred from, the health facilities visited, reasons for the choice of facility, treatments given at home and at the health facility visited, and sources of delay. Data on the duration of travel and the condition of the neonate on arrival at UWRH and SJH have been presented elsewhere. Demographic, clinical features, and social factors, such as age, sex, presenting features, diagnosis and outcome of the neonate, were also obtained. The information was obtained by trained nurses who worked in the hospitals. Some lived in the districts and could speak the dialects.

### 2.6. Data Analysis

The data were captured and analyzed using the Statistical Package for Social Sciences (SPSS) version 16.0 and cleaned by using standardized queries to conduct range and logic checks. Discrepant entries were identified and corrected. Frequencies, proportions, and means of study variables were computed and presented in tabular and graphical form. Comparisons and statistical inference were made using the Chi square test to assess risk factors, and the *t*-test was used to assess the degree of statistical significance when comparing means. Statistical significance was accepted at a 5% probability level, that is, a *p*-value of less than 0.05. Logistic regression was used to determine factors independently associated with delay in seeking health care for sick neonates. Caregivers were considered to have delayed care-seeking if they report to the neonatal unit after 24 h of onset of symptoms.

### 2.7. Ethical Clearance

Ethical clearance to perform this study was obtained from the Ghana Health Service Ethical Review Committee (Ethical Review Committee Protocol ID No: GHS-ERC 09/03/17). Consent was obtained before the questionnaires were administered. Permission was obtained from the facilities involved. The data were anonymized in order to not reveal patients’ identities, and the analysis was conducted in a way that the final results cannot be linked to individual patients.

## 3. Results

### 3.1. Characteristics of Patients and Caregivers

Altogether, the data were obtained from 153 caregivers of neonates that were referred to both hospitals, comprising 140 caregivers from the Upper West Regional Hospital and 13 caregivers of neonates referred to the St. Joseph’s Hospital. Mothers provided the information for 145 neonates. The ages of the mothers ranged from 17 to 48 years. More than a third, 39.9% (*n* = 61), had received basic education up to Junior High School or no education, whereas 92 (60.1%) had received Senior High School or a tertiary level education. Teenagers formed 12.5% (19) and 36.8% (7) of these teenage mothers that lost their babies at the hospitals. Only 3 out of the 19 teenagers reached the appropriate maximum schooling level of Senior High School (SHS). Four (4) of the teenagers were married, including two (2) out of the seven who lost their babies, confirming the occurrence of early marriage in this setting. The median age of the neonates was 7.5 days and their ages ranged from the first few hours of birth to 27 days. The majority of the neonates were seven or more days old at the time of hospital admission, 84 (54.9%). There were 88 (57.5%) males and 65 (42.5%) females translating into a sex ratio of 135 males to 100 females. There were no extremely low birth weight infants and only one very low birth weight infant. Premature babies comprised of 17.0% (26). Home visits were reported by 21.9% (32) of mothers, and 57.1% (16) of these visits were done by community health nurses, 39.3% (11) were done by community health volunteers, 3.6% (1) were done by a midwifes, and two did not specify the category of health staff who performed home visits. Table 1 provides a summary of the characteristics of the neonates that were referred to the hospital and their mothers.

### 3.2. Danger Signs and Home Treatments

Table 2 provides a summary of the reasons for referral. Fever, poor feeding, excessive crying, fast breathing, neonatal jaundice, abdominal distension, bleeding and skin lesions were the most common reasons for referral reported by the mothers. Fever was reported by 67 (44.1%) of the patients on arrival at the health facility, and 60 (93.8%) reporting fever had a temperature above 37.5 degree Celsius on arrival, whereas 4 (6.3%) of those who did not report fever had fever on arrival. Of those who died, two patients each presented symptoms of fever, noisy breathing, abdominal distension, poor feeding, and bleeding, whereas one patient presented symptoms of difficulty in breathing, fast breathing hypoglycemia, and altered sensorium.

The majority of caregivers did not give any home treatments before seeking care (60.8%, *n* = 93). Among those who administered orthodox medicine, 60 (39.2%) mothers gave some form of home based treatment to their neonates before seeking care, and paracetamol (acetaminophen) was the commonest medication that was given accounting for 40% (24). Inappropriate practices related to cord care, such as the use of cow dung, herbal baths, and treating babies with an enema, were also found (Table 3). Three (3) children who had these home treatments died; one applied toothpaste to the cord, another was given a herbal bath, and the third patient was treated with an enema. Figure 2 depicts the duration of illness before seeking care.

There were two peaks in the way care was sought. The first peak was on the first day of an illness (33.3% (*n* = 50)) and the second peak was between the 7th and 13th day of an illness (35.3% (*n* = 53)). Altogether, 67 (44.7%) caregivers sought care after 7 days of illness. The caregivers of two (16.7%) of the male neonates who died, sought care in less than 48 h, and one (8.3%) in the first week. Among the females, one (11.1%) sought care within 24 h of delivery, 3 (33.3%) sought care in the first week, and another 5 (55.6%) sought care in the second week of life as shown in Table 4.

### 3.3. Health Facilities Attended and Treatments Given

The majority of the caregivers sought care in a health facility as shown in Figure 3. Proximity, affordability, and the quality of care were the main determinants of choice of the health facility (Figure 4). More than half (61.4% (*n* = 94)) of the primary health care facilities visited referred patients to destination hospitals without giving any care themselves The reasons provided for not choosing a health facility in the first instance were the lack of a National Health Insurance Scheme (NHIS) coverage (1), the facility being too far (2), and other reasons reported by three caregivers.

### 3.4. Difficulties Experienced

Difficulties reported by caregivers, which may have affected their care seeking behaviors, were cultural barriers, such as prohibition of travelling with a baby, 5 (3.3%), and mothers being compelled to stay indoors for four to six weeks after the birth of a new baby, 6 (3.9%), anxiety and stress, 24 (15.6%), and fear of the hospital environment, 10 (6.5%). Others were “mother also not feeling well”, “my husband was not around to help me”, “nobody to take care of the other children”, and “the baby was too small and so I did not want to waste my time and still lose the baby”. Table 5 shows the results of the logistic regression of factors associated with the delay in seeking care.

Adjusting for maternal educational and marital status, neonates who received a home based remedy were 9.0 (95% CI = 2.9–27.5) times more likely to delay in seeking care at the neonatal unit compared to those who did not receive any home based remedy. Moreover, neonates with a low birth weight were less likely to delay in seeking care compared to normal or macrocosmic babies (Adjusted Odds Ratio (AOR) = 0.2, 95%CI = 0.1–0.4). Other significant factors associated with a delay in seeking care included increasing maternal age, no fever, and a normal breathing pattern of neonates.

### 3.5. Outcome

Twelve, 12 (7.8%), out of the 153 neonates died. There was a statistically significant difference between the mortality experience of male and female outborn neonates (X^2^, *p* = 0.018) such that female neonate were more likely to die (13.8%, 9/65) compared to male neonates (3.4%, 3/88). The three males who died suffered from meconium aspiration, birth asphyxia, and bleeding circumcision, respectively, whereas the females died mainly from communicable disease, such as sepsis, prematurity and its complications, as well as birth asphyxia. The mortality rate among the neonates over the study period was lower in UWRH (7.1% (*n* = 10)) compared to SJH (15.4% (*n* = 2)). However, there was no statistically significant difference in the mortality experience of outborn neonates admitted to the two facilities (chi square *p*-value = 0.290). Table 5 displays the factors independently associated with delays in seeking care at neonatal units.

## 4. Discussion

There were more male patients in this study than female patients. The predominance of male admissions is consistent with findings from other studies in similar settings and has been linked to the greater vulnerability of the male infant to illness and better care seeking for male children resulting from a preference for males [1,4,5,15,16]. In our study setting, most ethnic groups value male children more than female children and this may be the reason for the observed significant difference in mortality experienced between male and female neonates. We found no extremely low birth weight (ELBW) infants and only one very low birth weight (VLBW) neonate in this cohort of infants. Furthermore, low birth weight was associated with a delay in seeking health care. These findings could be an indication that ELBW and VLBW patients may have been unstable and therefore could not be stabilized before a referral or the referrals were not completed [14]. This is supported by a study at Kintampo, Ghana, which revealed an absence of care seeking for babies with a birth weight of 1.50–1.99 kg (AOR, 1.46) compared with infants who did not have a low birth weight [17]. In contrast, studies in Uganda, India, South Africa, and Egypt reported admissions of ELBW infants and a higher proportion of VLBW infants than this study [16,18,19,20]. Further studies are needed to identify the determinants of health seeking for ELBW and VLBW infants born in community settings, referral completion rates, and how to improve the confidence of caregivers in seeking medical care for small babies.

Only about a third of the patients, 50 (32.9%), were admitted in the first three (3) days of life, which is in contrast to a study from Nigeria which reported that the mean age of presentation of outborn neonates was 3.13 days in their study [21]. Altogether, 70 out of 153 outborn neonates (45.8%) were admitted within the first week of life whereas the majority, 82 (54.0%), were admitted in the second week of life and beyond. What is worrying is the observation that as many as 67 (44%) made the decision to seek care after their babies were sick for seven (7) or more days. This is inappropriate and it puts these babies at risk of severe disease with complications leading to death. It also suggests that even though only six patients admitted that the tradition of observing a period of seclusion of neonates for a week or more affected their decision to seek care, the practice was more widespread. This practice is not peculiar to Ghana and is widespread across Africa and Asian countries, such as Pakistan [4,22,23]. One of the ways to address this problem is through postnatal care to educate mothers on the practice and identify ill babies [24]. However, only about a fifth of these babies had some form of postnatal care. Thus, additional measures to scale up postnatal care and to curb this practice are necessary to avoid delays and reduce mortality rates to meet the SDG target.

Neonates with danger signs, such as fever, distended abdomen, and difficulty in breathing, were more likely not to delay reaching the neonatal unit. Most parents were able to recognize danger sign symptoms as other studies have also demonstrated [8,11,14,25,26]. All these signs can be associated with an infection; although, bleeding in newborns was most commonly caused due to hemorrhagic disease of the newborn [27,28]. Fever has been reported as a common complaint in other studies [8,10,11,25]. Having no fever and normal breathing were associated with a delayed admission in this study, suggesting that fever and abnormal breathing were readily observed, whereas other danger signs were not easily observed leading to a delay. Neonatal sepsis is one of the commonest causes of fever in neonates, although the contribution of malaria to neonatal disease in this endemic region needs to be further examined as neonatal malaria has been reported in other studies [16,18,21]. More effort to prevent infection in the community setting is needed.

The health center was the first place of seeking care for the majority, 87 (56.8%), of neonates followed by the CHPS compound, 29 (19%), the most basic health facility, while hospitals were accessible to only 18 (11.8%). It is reassuring to note that the majority of the patients sought care in a health facility and only a few patients sought care from an alternate source. This is in contrast to other studies which showed that most mothers preferred to visit untrained health providers [11,12,23]. A study in Ghana also showed that while 61% of caregivers sought care for sick neonates, only 39% sought the care of a doctor [29]. A study from Nigeria also showed that 5 out of 10 neonates did not seek any care for their children [25].

Proximity, the main determinant of the choice of a health facility, has been reported as an important determinant of care by other studies [7,10,12,23,30]. However, in spite of visiting a health facility, the majority of these outborns neonates did not receive any treatment prior to the referral. The finding is similar to a Tanzanian study, which reported that neonates were seen after multiple referrals, with one baby being referred five times. A study from Nigeria reported similar findings [25,31]. Reasons why smaller facilities in the Upper West Region of Ghana would refer without giving any care themselves have been identified as staff shortage and lack of adequate capacity to manage neonatal conditions at smaller facilities [32]. This highlights the need for capacity building to upgrade the level of neonatal care provided by primary care and a role for neonatal nurse practitioners to complement physician assistants in health centers [33,34]. Clear guidelines on the services each level of care should provide are also needed in settings where infection is common, as well as prompt referrals with proper attention to neonatal transport [31,35,36,37]. In the long term, regionalization of perinatal care and establishing intermediate care facilities close to communities may be the way forward [36,37].

Cultural traditions and social factors influence neonatal morbidity and mortality globally [4,15,22,23,33,38,39]. We also encountered the use of enemas as home remedies reported by five caregivers, the use of herbal baths reported by six caregivers, and application of cow dung reported by three caregivers. Use of such remedies have been reported in other studies [26,39]. Time spent observing these remedies tended to cause a significant delay in health seeking for medical care. Mothers who were above 30 years were less likely to seek care for their babies. This may be because they might have had more children and more experience in child care as lower care seeking rates have been associated with a parity of ≥4 [9]. It could also have been because of financial constraints or care considerations for the other children.

Changing these practices will require health promotion and community mobilization [7,8,23,33,40]. The use of women groups in rural communities in Nepal and community health volunteers to spearhead behavior change has shown some successes [33,41,42]. In Ghana, the newborn stakeholder’s forum is using newborn champions nominated from within the communities to lead these changes [43]. Further studies to assess the effectiveness of these community-based strategies are warranted to determine and disseminate the best practice. Some practices can also be tackled by substituting them with more baby friendly practices, such as using antiseptic baths to replace the herbal baths when more evidence regarding the safety of this intervention becomes available [44]. Postnatal care remains an important avenue for tackling these issues [24,45]. Other challenges mothers experienced when their babies were referred included anxiety and fears for which they required support. The WHO standards for improving neonatal care quality statements 6.2 calls for women to be supported emotionally to improve their mental health during childbirth, to give them a positive outlook, and improve the overall health outcomes of the mother and baby [13]. Interrogating these fears and stresses will optimize their biological, psychological, social, and cultural well-being.

There were some limitations to this study. The study duration was short due to limited time, and the funding provision was to finish the study within a year. We were also only able to include 13 patients from Jirapa because there were fewer outborn patients there during the initial study period, and we were unable to continue the data collection there after an extension period was granted for the study. It is possible that patients in this study were more likely to seek care in a health facility, being a facility-based study. Thus, further studies from the community are required to provide a full picture.

## 5. Conclusions

Socio-cultural traditions relating to teenage pregnancy, early marriage, cord care, herbal baths, and a period of confinement still influence health practices, outcomes, and health seeking behavior in this setting. Therefore, while efforts are being made to improve facility-based neonatal care, commensurate efforts to educate and improve care in the community should occur concurrently. A fifth of the neonates in this study received postnatal care, and danger signs, especially obvious ones like fever, were identified. Since danger signs pertaining to infection where most common efforts to reduce infection through community related activities are needed, further education on danger signs that are not so readily observed are needed as well as. While the majority of caregivers sought care at a health facility, there appeared to be a limited repertoire of interventions for treating neonatal diseases in the primary care settings as the majority of patients were referred without any prior treatment. Improving access to postnatal care, bringing specialized care closer to communities through the training of neonatal nurse practitioners, education, and advocacy using multidisciplinary approaches are needed to mitigate these problems. A larger community-based study can further explore solutions for these findings, and provide information on referral completion. Furthermore, the larger community-based study may focus in-depth on identifying specific community and provider-level barriers to resolve these problems, including referral and pre-referral treatment.

## Figures and Tables

**Figure 1 children-07-00015-f001:**
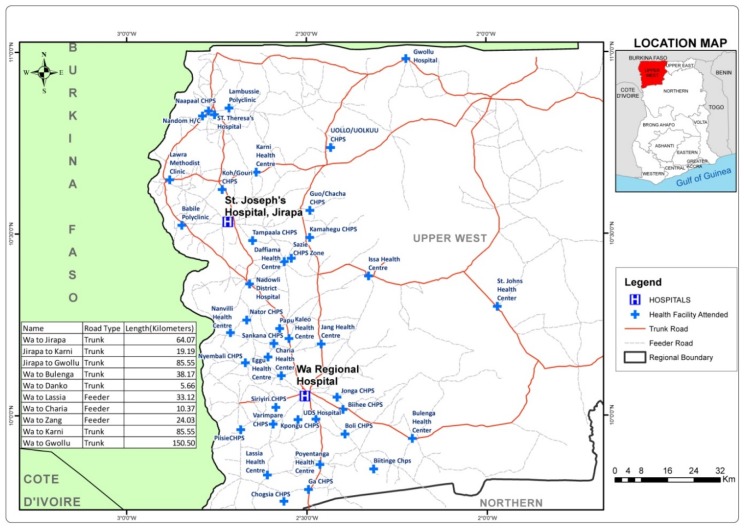
Health facilities attended, St. Joseph’s Hospital and Upper West Regional Hospital, in Wa, Ghana.

**Figure 2 children-07-00015-f002:**
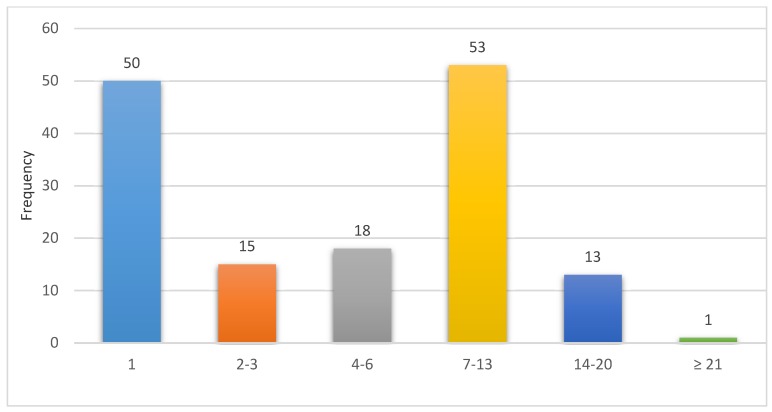
Duration of illness before seeking care.

**Figure 3 children-07-00015-f003:**
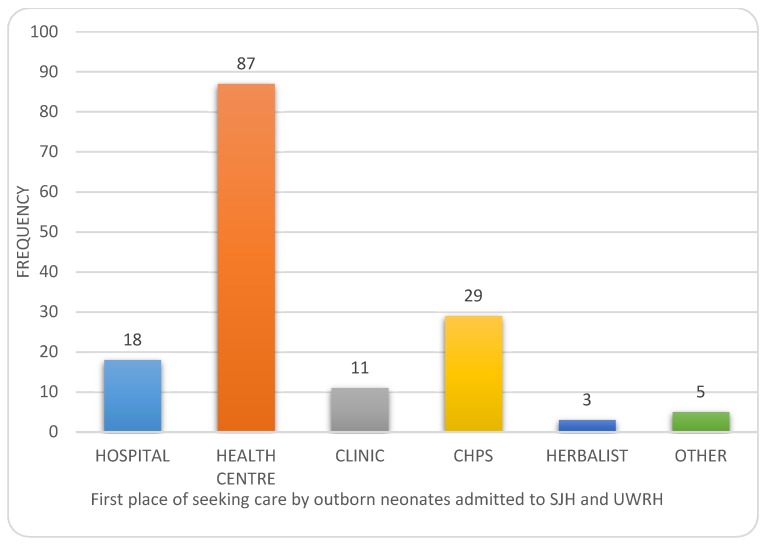
The first place of seeking care by outborn neonates.

**Figure 4 children-07-00015-f004:**
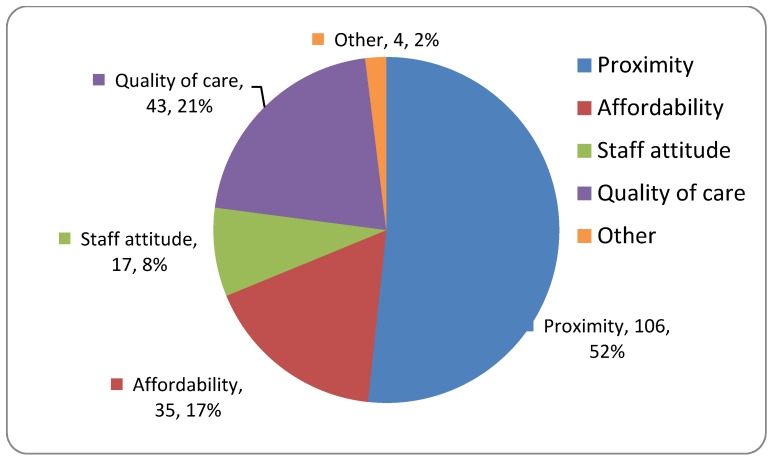
Reasons given by caregivers of outborn neonates for choosing a health facility.

**Table 1 children-07-00015-t001:** Characteristics of outborn neonates and their mothers.

	Frequency	Percent (%)
**MATERNAL CHARACTERISTICS**		
**Maternal Age (years)**		
<20	19	12.4
20–35	115	75.2
>35	19	12.4
**Educational Status**		
None	14	9.2
Primary	27	17.6
JHS	20	13.1
SHS	55	35.9
Tertiary	37	24.2
**Marital Status**		
Married	118	77.1
Divorced	2	1.3
Cohabitation	2	1.3
Single/Never married	31	20.3
**Religion**		
Christian	77	50.3
Moslem	74	48.4
Traditionalist	2	1.3
**NEONATAL CHARACTERISTICS**		
**Age (days)**		
≤1	30	19.6
2–3	19	12.4
4–6	20	13.1
7–14	61	39.9
≥15	23	15.0
**Sex**		
Male	88	57.5
Female	65	42.5
**Birth weight (grams)**		
<1000	0	0.0
1000–1499	1	0.7
1500–1999	8	5.2
2000–2499	12	7.8
≥2500	132	86.3
**Gestational age**		
<37 weeks	26	17.0
≥37 weeks	127	83.0
**Place of admission**		
Upper West R. Hospital	140	91.5
St. Joseph’s Hospital	13	8.5

**Table 2 children-07-00015-t002:** Reasons for the referral to Upper West Regional Hospital (UWRH) and St. Joseph’s Hospital (SJH) (multiple answers apply).

	Reasons for the Referral	Frequency	Percentage (%)
1	Fever	67	44.1
2	Breathing abnormalities	35	23.5
3	Excessive crying	27	17.8
4	Poor feeding	24	15.8
5	Abdominal Distension	17	11.2
6	Jaundice	16	10.5
7	Bleeding	13	8.6
8	Vomiting	12	7.9
9	Skin lesions	10	6.6
10	Prematurity	8	5.2
11	Eye discharge	8	5.2
12	Discharging/weak cord	7	4.6
13	Low birth weight	7	4.6
14	Cough	5	3.3
15	Febrile convulsion	4	2.6
16	Deformity/malformation	4	2.6
17	Impetigo	3	2.0
18	Congenital malformations	3	2.0
19	Inability to pass or difficulty to passing stool	3	2.0
20	Diarrhea	2	1.3
21	Hypoglycemia	2	1.3
22	Hypothermia	2	1.3
23	Altered sensorium	1	0.7
24	Non-febrile convulsion	1	0.7
	* Others	9	5.9

* Other: Each baby suffered from one of these illnesses: breast abscess, big baby, runny nose, loose stools, shoulder fracture, no breathing pattern, inability to cry, severe birth asphyxia, and swelling on the head.

**Table 3 children-07-00015-t003:** Treatments received by outborn infants prior to reaching referral hospital.

Treatment (Multiple Response)	Frequency	Percentage
**Home treatments**		
Medicine to drink		
Herbal medicine	7	4.5
Paracetamol syrup	21	13.5
Teedar (paracetamol + diphenhydramine) syrup	3	1.9
Application of agents to cord		
Breast milk	2	1.3
Applied toothpaste to cord	1	0.6
Used powder	3	1.9
Used cow dung	3	1.9
Shea butter	2	1.3
Palm kernel oil	1	0.6
Other Herbal applications		
Herbal enema	5	3.2
Herbal baths	6	3.8
Herbal lotions/creams	2	1.3
Substances applied to eyes		
Salt water	4	2.6
Breast milk	1	0.6
No home therapy		
Did nothing	93	60.8
**Treatment at referring health facility prior to referral to neonatal unit**		
Antibiotics were given	2	1.3
Cord was dressed	4	2.6
Paracetamol syrup was given	9	5.9
Salvon was given to baby	1	0.7
Tepid sponging	1	0.7
Baby was detained, medication given and referred afterwards	30	19.6
Nothing was done	2	1.3
Not applicable	6	3.9
Referral note was given	94	61.4
Other	4	2.6

**Table 4 children-07-00015-t004:** Duration of illness and mortality of outborn neonates by sex.

Duration of Illness Before Seeking Care	SEX		
	Male	Female	Total
<24 h	0 (0)	1 (11.1%)	1 (8.3)
1st week	2 (66.7%)	3 (33.3%)	5 (41.7)
2nd week	1 (33.3%)	5 (55.6%)	6 (50.0)

**Table 5 children-07-00015-t005:** Logistic regression of factors independently associated with delay in seeking care at a neonatal unit.

Characteristics	OR (95% CI)	*p*-Values	AOR * (95% CI)	*p*-Values
Birth weight				
Low birth weight	0.2 (0.1–0.4)	0.000	0.2 (0.1–0.4)	0.000
Normal or macrosomia	-			
Home based remedy administration				
Home remedy administered	8.5 (2.8–25.3)	0.000	9.0 (2.9–27.5)	0.000
Home remedy not administered	-			
Neonate fever				
No Fever	6.5 (2.5–16.6)	0.000	6.7 (2.6–17.6)	0.000
Had Fever	-			
Breathing pattern				
Normal breathing	6.6 (2.0–20.8)	0.002	6.5 (2.0–21.1)	0.002
Abnormal breathing	-			
Went to postnatal clinic/health center				
Less than three times	-			
More than four times	2.4 (1.0–5.6)	0.050	2.7 (1.1–7.1)	0.040
Maternal Age				
20 years and below	-			
21–30 years	1.8 (0.7–4.8)	0.2	2.5 (0.7–9.0)	0.151
More than 30 years	2.9 (0.7–3.6)	0.06	4.4 (1.0–4.7)	0.045

AOR *—Adjusted Odds Ratio, adjusting for educational status, marital status.

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
