# Peer review of "The Profile, Health Seeking Behavior, Referral Patterns, and Outcome of Outborn Neonates Admitted to a District and Regional Hospital in the Upper West Region of Ghana: A Cross-Sectional Study"

_children, 2020, doi:10.3390/children7020015_

Round 1

Reviewer 1 Report

The profile, health seeking behaviour, referral patterns and outcome of outborn neonates admitted to a district and regional hospital in the Upper West Region of Ghana; a cross sectional study.

This is a very interesting and useful paper.  There are several areas, however, where the manuscript can be strengthened.

P2 L62: Please briefly indicate the proximity of the two hospitals to each other, and whether there are any reasons other than distance that a patient would go to one or the other, in terms of available services or other factors.

P2 L85-87: Please explain your purposive sampling – how was the eligible sample determined?  What was the “purposive” aspect?  Also please clarify the reference to low numbers – of births?  of study participation?

P2 L84: The sample size calculations are confusing.  Only later in the manuscript do you clarify that the total candidate population was from both hospitals, but not whether all neonates were considered or only those admitted for illness.  Please provide a basis (reference) for your assumptions about percentage of newborns requiring admission and percentage out-born, and also define the term out-born.

P3 L103: Please provide a reference for the information on duration of travel “presented elsewhere.”  It would be useful to include a sentence clarifying the essential difference between the current and present manuscript.

P3 L126: Your analysis did not describe whether there were overall differences in the experiences of the 13 from St Joseph’s and those from UWRH.  It may be better practice to limit the analysis to UWRH.  Even though the St Joseph’s cases put you over your goal, the UWRH numbers are still more than your calculated minimum sample size. 

P3 L130: Did your calculations for education take into account that teenagers may have been too young to have completed all the grades?  There are ways to incorporate the concept of “appropriate/maximum schooling given age.”

P5 L143: This section would be easier to interpret if it were broken into three paragraphs: referral reasons, home treatments and delays.  All three paragraphs should be organized more in line with their tables/figure, which are more straight-forward.  Please provide the percentage of caregivers who did not give home treatments at the end of this sentence (P5 L152).

P5 L158: Only one value can be the majority.

P5 L161: Were there male vs. female differences in the patterns of delay, particularly the extreme delay?  This is referred to later on, but should be provided here.

P7 L168: This section is another instance where there may be differences between the two institutions.  If St Joseph’s is retained, this needs to be investigated and (probably briefly) mentioned.  Is there any information on the distance from the various care options to the hospitals?  Or any references/information on why the smaller facilities would refer without giving any care themselves?

P7 L172: The “94 ( 61.4%)” seems out of place.

P8 L185: The logistic regression results would seem to be a main outcome of your paper, and thus should be summarized here.

P8 L188-196: The male/female difference is intriguing but confusing.  This discussion in this section needs to be better organized.  The discussion in P9 L205 could include this concept.

P9 L208-211: There are too many ideas in this one sentence.  The reference to short duration of the study and sampling process needs to have been established back on page 2.  The discussion of unstable ELBW/VLBW should be tied to the observation of only 1 such infant participating in your study.  The Bangladesh reference (#12) is appropriate but not needed.

P10 L234-247: This paragraph largely repeats information already given and is not really needed.

P10 L255: It is actually likely that since your study was derived from patients who reached a hospital, that your respondents are a biased subset of the community (that’s ok, just needs to be acknowledged).  This should be in the limitations.

P10 L269-271: You did not have many teens and did not provide much information on them, so these sentences are true but not relevant to the focus of your paper.  P10 L274 “Time spent…” is the more important concept.

P11 L288-296: Part of this paragraph belongs in the previous section; Limitations should be a separate paragraph.

P11 L311: It would be worthwhile for the “larger community-based study” to focus in-depth on identifying specific community- *and* provider-level barriers to resolving these problems, including referral and pre-referral treatment, in addition to exploring solutions.

Author Response

Reviewer 1

This is a very interesting and useful paper.  There are several areas, however, where the manuscript can be strengthened.

Response: Thank you for your kind review of the manuscript and offering suggestions to improve upon the manuscript

P2 L62: Please briefly indicate the proximity of the two hospitals to each other, and whether there are any reasons other than distance that a patient would go to one or the other, in terms of available services or other factors.

 Response: This suggestion was carried out. A map of the area and a phrase has been added to address this suggestion. Please refer P2L 77-84

P2 L85-87: Please explain your purposive sampling – how was the eligible sample determined?  What was the “purposive” aspect?  Also please clarify the reference to low numbers – of births?  of study participation?

 Response: The sentences were rewritten to reflect what we did and also to give better clarity. The following was inserted;

“Parents or caregivers of all the 153 consecutive outborn neonates admitted over the period of the study were interviewed. Although the study collected data over an eight (8) month period from 23rd January to 16th October 2018, the last outborn neonate from SJH was admitted in June 2019 resulting in the low number of outborn neonates from SJH. P3L92-95

P2 L84: The sample size calculations are confusing.  Only later in the manuscript do you clarify that the total candidate population was from both hospitals, but not whether all neonates were considered or only those admitted for illness.  Please provide a basis (reference) for your assumptions about percentage of newborns requiring admission and percentage out-born, and also define the term out-born.

 Response: This was clarified and Outborn was defined. The section was re-written to give a clearer meaning as found in P3L95-101.

P3 L103: Please provide a reference for the information on duration of travel “presented elsewhere.”  It would be useful to include a sentence clarifying the essential difference between the current and present manuscript.

 Response: The manuscript referred to is under peer review by same journal and titled “The transport and outcome of sick outborn neonates admitted to a regional and district hospital in the Upper West Region of Ghana: a cross-sectional study” It focussed on the transport conditions, distance and duration of travel of outborn neonates to the referral facility and their condition on arrival.

P3 L126: Your analysis did not describe whether there were overall differences in the experiences of the 13 from St Joseph’s and those from UWRH.  It may be better practice to limit the analysis to UWRH.  Even though the St Joseph’s cases put you over your goal, the UWRH numbers are still more than your calculated minimum sample size. 

 Response: The goal was to cover all outborn neonates admitted to the two facilities during the period of the study. A sentence was added to the outcome section to address the suggestion on experience. P9, L208-210. It reads

“The mortality rate among the neonates over the period was lower in UWRH [7.1% (n=10)] compared to SJH [15.4% (n=2)]. However, there were no statistically significant difference in the mortality experience of neonates admitted to the two facilities (chi square p-value=0.290).”

P3 L130: Did your calculations for education take into account that teenagers may have been too young to have completed all the grades?  There are ways to incorporate the concept of “appropriate/maximum schooling given age.”

 Response: No please. Looking at the data again, there were 19 teenagers in the study aged 17-19. Out of this, only three had reached the appropriate maximum schooling level of SHS. The rest either  P4, L 143-144

P5 L143: This section would be easier to interpret if it were broken into three paragraphs: referral reasons, home treatments and delays.  All three paragraphs should be organized more in line with their tables/figure, which are more straight-forward.  Please provide the percentage of caregivers who did not give home treatments at the end of this sentence (P5 L152).

 Response: Thank you, this suggestion was carried out. P5 L157. Also the percentage of caregivers who did not give home treatment was stated as [60.8%, n=93]. P6, L165

P5 L158: Only one value can be the majority.

 Response: Thank you for pointing out this. It was rectified and the sentence now reads There were two peaks in the way care was sought, the first peak was on the first day of illness [33.3% (n= 50)] and the second peak was between the 7th and 13th day of illness [35.3% (n= 53)]. P6 L173-174

P5 L161: Were there male vs. female differences in the patterns of delay, particularly the extreme delay?  This is referred to later on, but should be provided here.

 Response: This suggestion was carried out and the sentence below was moved to P6,L176-179 as suggested.

“The caregivers of two (16.7%) of the male neonates who died, sought care in less than 48 hours, and one (8.3%) in the first week; among the females, one (11.1%) sought care within 24 hours of delivery, 3(33.3%) sought care in the first week and another 5 (55.6%) sought care in the second week of life as shown in table 4.”

P7 L168: This section is another instance where there may be differences between the two institutions.  If St Joseph’s is retained, this needs to be investigated and (probably briefly) mentioned.  Is there any information on the distance from the various care options to the hospitals?  Or any references/information on why the smaller facilities would refer without giving any care themselves?

 Response: St Joseph’s hospital was retained and the difference in mortality experience among the neonates stated in P9L210-213. A map of the study area showing the health facilities, and distance from various health care options including the referral neonatal units was inserted under the study area section and labelled figure 1.

Reference information on why the smaller facilities would refer without giving any care themselves was added to the discussion, P11, L283-285 as shown below. The source of the information cited accordingly.

“Reasons why smaller facilities in the upper West region of Ghana would refer without giving any care themselves have been identified as staff shortage and lack of adequate capacity to manage neonatal conditions at the smaller facilities”.

P7 L172: The “94 ( 61.4%)” seems out of place.

 Response: It was placed in the appropriate place as shown in P8, L189

P8 L185: The logistic regression results would seem to be a main outcome of your paper, and thus should be summarized here.

 Response: A summary of the logistic regression findings were included as suggested P9L205-2010

“Adjusting for maternal educational and marital status, neonates who received home based remedy were 9.0 (95% CI= 2.9-27.5) times more likely to delay in seeking care at the neonatal unit compared to those who did not receive any home based remedy. Also neonates with low birth weight were less likely to delay in seeking care compared to normal or macrocosmic babies [AOR=0.2, 95%CI=0.1-0.4]. Other significant factors associated with delay in seeking care include increasing maternal age, no fever and normal breathing pattern of neonates”

P8 L188-196: The male/female difference is intriguing but confusing.  This discussion in this section needs to be better organized.  The discussion in P9 L205 could include this concept.

 Response: This section was reorganised to convey the findings as suggested. P9L213-217. The discussion also structured to well contain the concept as suggested. P10 L230-233

P9 L208-211: There are too many ideas in this one sentence.  The reference to short duration of the study and sampling process needs to have been established back on page 2.  The discussion of unstable ELBW/VLBW should be tied to the observation of only 1 such infant participating in your study.  The Bangladesh reference (#12) is appropriate but not needed.

 Response: The long sentence in P10,L233 was rewritten into two sentences to make the message clearer. The reference to short duration was consequently taken the limitation section on P12, L316-317. Also the Bangladesh reference was also expunged from this section as suggested

P10 L234-247: This paragraph largely repeats information already given and is not really needed.

 Response: The repeated information was taken out and the relevant information maintained

P10 L255: It is actually likely that since your study was derived from patients who reached a hospital, that your respondents are a biased subset of the community (that’s ok, just needs to be acknowledged).  This should be in the limitations.

 Response: The statement was moved to the limitation section as suggested

P10 L269-271: You did not have many teens and did not provide much information on them, so these sentences are true but not relevant to the focus of your paper.  P10 L274 “Time spent…” is the more important concept.

 Response: The sentence on teenage pregnancy was taken away as suggested P11, L290

P11 L288-296: Part of this paragraph belongs in the previous section; Limitations should be a separate paragraph.

 Response: Thank you, suggestions carried out, P12, L307-312

P11 L311: It would be worthwhile for the “larger community-based study” to focus in-depth on identifying specific community- *and* provider-level barriers to resolving these problems, including referral and pre-referral treatment, in addition to exploring solutions

Response: Your kindly suggested recommendation was incorporated into our recommendation.  P12, L335

Thank you for significantly improving upon the equality of the manuscript. 

Reviewer 2 Report

Thank you very much for submitting “The profile, health seeking behavior, referral patterns and outcome of outborn neonates admitted to a district and regional hospital in the Upper West Region of Ghana; a cross-sectional study” for consideration. This is a cross-sectional study examining the barriers to seeking medical attention in Ghana for neonates that were admitted to a regional hospital and born outside of the hospital. This study addresses a major health concern, as newborn health is a major contributor to mortality of children under the age of 5 in low and middle income countries. This study identifies common demographics, reasons for referral and home remedies associated with outborn neonates admitted to a regional hospital and those that died. Fever, breathing abnormalities abdominal distension, bleeding and poor feeding were associated with death. Poor cord care was found in children that died. Birth asphyxia, aspiration, bleeding circumcision, sepsis, and prematurity were the leading diagnosis at death. Unique findings were that many sought care after 7 days of illness, and this study supported previous findings that families with male babies presented more frequently than female children. This study brings good evidence to the need of post-natal care and education to families in rural Ghana. There are good acknowledgements of the biases of the paper (such as being from a health facility and missing those that may seek alternate treatment or no treatment). This is a very good study on an excellent topic addressing a major global health concern.

Author Response

Reviewer 2

Thank you very much for submitting “The profile, health seeking behavior, referral patterns and outcome of outborn neonates admitted to a district and regional hospital in the Upper West Region of Ghana; a cross-sectional study” for consideration. This is a cross-sectional study examining the barriers to seeking medical attention in Ghana for neonates that were admitted to a regional hospital and born outside of the hospital. This study addresses a major health concern, as newborn health is a major contributor to mortality of children under the age of 5 in low and middle income countries. This study identifies common demographics, reasons for referral and home remedies associated with outborn neonates admitted to a regional hospital and those that died. Fever, breathing abnormalities abdominal distension, bleeding and poor feeding were associated with death. Poor cord care was found in children that died. Birth asphyxia, aspiration, bleeding circumcision, sepsis, and prematurity were the leading diagnosis at death. Unique findings were that many sought care after 7 days of illness, and this study supported previous findings that families with male babies presented more frequently than female children. This study brings good evidence to the need of post-natal care and education to families in rural Ghana. There are good acknowledgements of the biases of the paper (such as being from a health facility and missing those that may seek alternate treatment or no treatment). This is a very good study on an excellent topic addressing a major global health concern

Response:

Thank you very much for your review and kind words.